# LOOKBEHIND-SAM: $k$ STEPS BACK, 1 STEP FORWARD

## ABSTRACT

Sharpness-aware minimization (SAM) methods have gained increasing popularity by formulating the problem of minimizing both loss value and loss sharpness as a minimax objective. In this work, we increase the efficiency of the maximization and minimization parts of SAM's objective to achieve a better loss-sharpness trade-off. By taking inspiration from the Lookahead optimizer, which uses multiple descent steps ahead, we propose Lookbehind, which performs multiple ascent steps behind to enhance the maximization step of SAM and find a worst-case perturbation with higher loss. Then, to mitigate the variance in the descent step arising from the gathered gradients across the multiple ascent steps, we employ linear interpolation to refine the minimization step. Lookbehind leads to a myriad of benefits across a variety of tasks. Particularly, we show increased generalization performance, greater robustness against noisy weights, as well as improved learning and less catastrophic forgetting in lifelong learning settings.

## 1 INTRODUCTION

Improving the optimization methods used in deep learning is a crucial step to enhance the performance of current models. Notably, building upon the long-recognized connection between the flatness of the loss landscape and generalization (Hochreiter & Schmidhuber, 1994; Keskar et al., 2016; Dziugaite & Roy, 2017; Neyshabur et al., 2017; Izmailov et al., 2018), sharpness-aware training methods have gained recent popularity due to their ability to significantly improve generalization performance compared to minimizing the empirical risk using stochastic gradient descent (SGD). Particularly, sharpness-aware minimization (SAM) (Foret et al., 2021) was recently proposed as an effective means to simultaneously minimize both loss value and loss sharpness during training. Given a neural network with parameters $\phi$, some loss function $L(\phi)$, SAM seeks parameters in flat regions by formulating the problem as a minimax optimization:

$$\min_{\phi} \max_{\|\epsilon\|_2 \leq \rho} L(\phi + \epsilon), \qquad (1)$$

where worst-case perturbations $\epsilon$ are applied to parameters $\phi$, with the distance between original and perturbed parameters being controlled by $\rho$. SAM approximates the maximization step by first performing a single gradient ascent step and then using the gradient of the loss to do a single descent step from the original solution. This leads to finding a low-loss parameter configuration $\phi$ such that the loss is also low in the neighborhood $\rho$ which will lead to flatter solutions. Several follow-up methods have emerged to further enhance its performance (Kwon et al., 2021; Zhuang et al., 2022; Kim et al., 2022) and reduce its computation overhead (Du et al., 2022a;b; Liu et al., 2022a).

Despite the recent success, improving upon SAM requires a delicate balance between loss value and sharpness. Ideally, the optimization process would converge towards minima that offer a favorable compromise between these two aspects, thereby leading to high generalization performance. However, naively increasing the neighborhood size $\rho$ used to find the perturbed solutions in SAM leads to a considerable increase in training loss, despite improving sharpness (Figure 1, full circles). In other words, putting too much emphasis on finding the worst-case perturbation is expected to bias convergence to flat but high-loss regions and negatively impact generalization performance.

Instead of performing a single ascent step akin to SAM, performing multiple ascent steps is a promising way of increasing the neighborhood region to find perturbed solutions, and thus further reducing sharpness. However, this is not what is observed empirically (Figure 1, empty circles). In fact, previous works (Foret et al., 2021; Andriushchenko & Flammarion, 2022) have shown that

such a multistep variant may hurt performance. A possible cause is the increased gradient instability originating from moving farther away from our original solution (Liu et al., 2022b). Note that such instability may also be present when using a high $\rho$, even in single-ascent step SAM. In this case, applying a variance reduction technique such as Lookahead (Zhang et al., 2019) with SAM as inner optimizer may help mitigate the performance loss when using larger $\rho$. However, as we demonstrate in our experiments, this is also not helpful (Figure 1, empty triangles).

In this work, we present a novel optimization method, called Lookbehind, that leverages the benefits of multiple ascent steps and variance reduction to improve the efficiency of the maximization and minimization parts of equation 1. This leads to Lookbehind successfully reducing both loss and sharpness across small and large neighborhood sizes (Figure 1, full triangles), achieving the best loss-sharpness trade-off.

In practice, improving the loss and sharpness trade-off results in a myriad of benefits across several training regimes. Particularly, when applying Lookbehind to SAM and ASAM, we show a considerable improvement in terms of generalization performance across several models and datasets. Moreover, models trained with Lookbehind have increased robustness against noisy weights at inference time. Lastly, we evaluate Lookbehind in the context of lifelong learning and show an improvement both in terms of learning and catastrophic forgetting on multiple models and datasets.

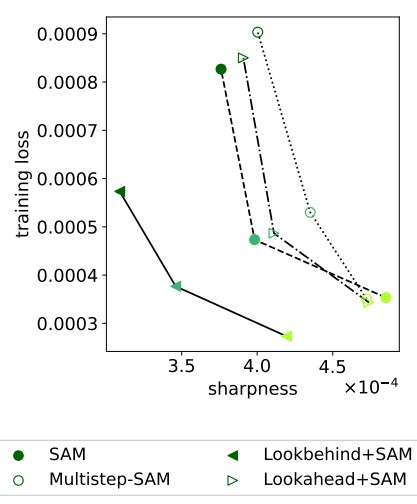

Figure 1: Loss and sharpness trade-off using ResNet-34 trained on CIFAR-10. Darker shades indicate training with higher neighborhood sizes $\rho \in \{0.05, 0.1, 0.2\}$. Lookbehind achieves both lower loss and sharpness.

## 2 BACKGROUND: SHARPNESS-AWARE MINIMIZATION

Our method, Lookbehind, builds upon sharpness-aware minimization (SAM) methods with the goal of solving the inner maximization problem of SAM more accurately while stabilizing the outer minimization part of SAM's objective. We will start by briefly introducing the sharpness-aware minimization methods used throughout the paper.

To solve the problem in equation 1 using standard stochastic gradient methods, SAM (Foret et al., 2021) proposes to estimate the gradient of the minimax objective in two steps. The first step is to approximate the inner maximization $\epsilon(\phi)$ using one step of gradient ascent; the second is to compute the loss gradient at the perturbed parameter $\phi + \epsilon(\phi)$. This leads to the following parameter update:

$$\phi_t = \phi_{t-1} - \eta \nabla_\phi L(\phi_{t-1} + \epsilon(\phi_{t-1})), \quad \epsilon(\phi) := \rho \frac{\nabla L(\phi)}{||\nabla L(\phi)||_2}. \tag{2}$$

Several follow-up sharpness-aware methods have been proposed to further improve upon the original formulation. Notably, a conceptual drawback of SAM is the use of a fixed-radius Euclidean ball as maximization neighborhood, which is sensitive to re-parametrizations such as weight re-scaling (Dinh et al., 2017; Stutz et al., 2021). To address this problem, ASAM (Kwon et al., 2021) was proposed as an adaptive version of SAM, which redefines the maximization neighborhood in equation 1 as component-wise normalized balls $\|\epsilon/|\phi|\|_2 \leq \rho$. This leads to the modified parameter update:

$$\phi_t = \phi_{t-1} - \eta \nabla_\phi L(\phi_{t-1} + \epsilon(\phi_{t-1})), \quad \epsilon(\phi) := \rho \frac{T_\phi^2(\nabla L(\phi))}{||T_\phi(\nabla L(\phi))||_2} \tag{3}$$

where $T_\phi(v) := \phi \odot v$ denotes the component-wise multiplication operator associated to $\phi$. In what follows, we use both SAM and ASAM as our baseline sharpness-based learning methods.

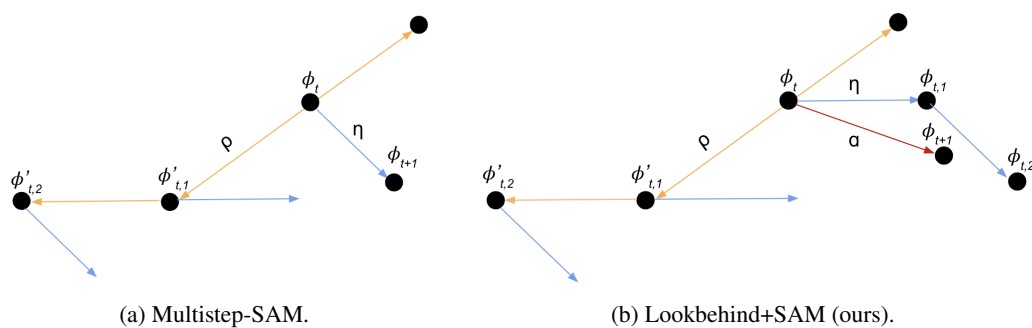

(a) Multistep-SAM.                    (b) Lookbehind+SAM (ours).

Figure 2: Illustration of Multistep-SAM (a) and Lookbehind-SAM (b) using $k = 2$.

## 3  LOOKBEHIND OPTIMIZER

Our algorithm, Lookbehind (+SAM), presents a novel way to improve the solution found by SAM's objective (equation 1). The intuition of Lookbehind is two-fold. First, we improve the maximization part of SAM's objective by performing multiple ascent steps to find a worst-case weight perturbation that has a higher loss than the original, single-step SAM within a given neighborhood of the original point. We refer to such maximization of the loss as we perform multiple ascent steps in SAM as looking behind. In other words, we are looking behind in the sense that we are climbing the loss landscape. (This term is inspired by the Lookahead optimizer (Zhang et al., 2019), where looking ahead refers to the minimization of the loss as they perform multiple descent steps.)

Second, to improve the minimization part of SAM's objective, we reduce the variance derived from the multiple ascent steps by aggregating the gradients along the way for the descent step and performing linear interpolation in the parameter space. This results in an alleviation of the instability that arises from (1) performing multiple ascent steps due to the various gradients gathered in the ascent phase not being aligned with each other and (2) the substantial departure away from the original point as performing ascent steps, which negatively impacts SAM's minimization objective and consequent loss-sharpness trade-off (Figure 1). Lookbehind combines instead the gradients computed at intermediate distances, improving upon the multiple ascent step variant of SAM (Multistep-SAM). A visual comparison between Multistep-SAM and Lookbehind is illustrated in Figure 2.

While Multistep-SAM performs $k$ ascent steps ($\phi'_{t,1}, \cdots, \phi'_{t,k}$) and uses the gradient from the last step ($\phi'_{t,k}$) for the final update, Lookbehind uses slow weights ($\phi_t, \phi_{t+1}, \cdots$) and fast weights ($\phi_{t,1}, \cdots, \phi_{t,k}$), where fast weights are updated using the gradients from $k$ ascent SAM steps. Then, the slow weights are updated toward the fast weights through linear interpolation. Even though both methods entail the same number of gradient computations, Lookbehind has a stabilizing effect over Multistep-SAM by combining the gradient information.

The pseudo-code for Lookbehind is presented in Algorithm 1. After synchronizing the fast weights (line 2) and the perturbed weights (line 3), we sample a minibatch (line 4) and perform $k$ ascent steps of SAM by preserving the previously perturbed slow weights (line 7) and introducing further perturbations in the subsequent inner step (line 6); corresponding descent steps are tracked and the fast weights are updated accordingly (line 8). After $k$ steps, a linear interpolation of the fast and slow weights is conducted (line 10).

---

**Algorithm 1** Lookbehind+SAM

**Require:** Parameters $\phi_0$, loss $L$, inner steps $k$, slow and fast weights step sizes $\alpha$ and $\eta$, neighborhood size $\rho$, training set $D$
1: **for** $t = 1, 2, \ldots$ **do**
2: $\quad \phi_{t,0} \leftarrow \phi_{t-1}$
3: $\quad \phi'_{t,0} \leftarrow \phi_{t-1}$
4: $\quad$ Sample mini-batch $d \sim D$
5: $\quad$ **for** $i = 1, 2, \ldots, k$ **do**
6: $\quad\quad \epsilon \leftarrow \rho \dfrac{\nabla L_d(\phi'_{t,i-1})}{\|\nabla L_d(\phi'_{t,i-1})\|_2}$
7: $\quad\quad \phi'_{t,i} \leftarrow \phi'_{t,i-1} + \epsilon$
8: $\quad\quad \phi_{t,i} \leftarrow \phi_{t,i-1} - \eta \nabla_{L_d}(\phi'_{t,i})$
9: $\quad$ **end for**
10: $\quad \phi_t \leftarrow \phi_{t-1} + \alpha(\phi_{t,k} - \phi_{t-1})$
11: **end for**
12: **return** $\phi$

---

## 4 EXPERIMENTAL RESULTS

In this section, we start by introducing our baselines (Section 4.1), and then we conduct several experiments to showcase the benefits of achieving a better sharpness-loss trade-off in SAM methods. Particularly, we test the generalization performance on several models and datasets (Section 4.2) and analyze the loss landscapes at the end of training in terms of sharpness (Section 4.3). Then, we study the robustness provided by the different methods in noisy weight settings (Section 4.4). Lastly, we analyze how the ability to continuously learn is affected in sequential training settings (Section 4.5).

For the following experiments, we use residual networks (ResNets) (He et al., 2016) and wide residual networks (WRN) (Zagoruyko & Komodakis, 2016) models trained from scratch on CIFAR-10, CIFAR-100 (Krizhevsky et al., 2009), and ImageNet (Deng et al., 2009). We report the mean and standard deviation over 3 different seeds throughout the paper unless noted otherwise. Additional training and hyperparameter search details are provided in Appendices A.3 and A.4.

### 4.1 BASELINES

On top of the previously discussed Lookbehind+SAM method, we note that our algorithm can be easily extended to ASAM by using the component-wise rescaling (equation 3) in the inner loop updates. We call this variant Lookbehind+ASAM. Additionally to SGD, vanilla SAM, and vanilla ASAM, we compare Lookbehind+SAM/ASAM to the following methods:

- *Multistep-SAM/ASAM*: As previously discussed in Section 3, this corresponds to performing multiple ascent steps to the vanilla SAM and ASAM algorithms, with the final update using the gradient from the last step.

- *Lookahead+SAM/ASAM*: We use Lookahead with sharpness-aware methods by applying single-step SAM and ASAM as the inner optimizers. A detailed description of Lookahead+SAM/ASAM is provided in Appendix A.2.

- *Lookahead+SGD*: For the sake of completeness, we also apply the Lookahead optimizer to SGD, as originally proposed by Zhang et al. (2019).

### 4.2 GENERALIZATION PERFORMANCE

We start by reporting the generalization performance on several models and datasets in Table 1. We observe that models trained with Lookbehind achieve the best generalization performance across all architectures and datasets. This is observed for both SAM and ASAM. Moreover, we see the Lookbehind+SAM/ASAM variants always outperform Lookahead+SGD, which further validates applying Lookbehind to sharpness-aware minimization methods. Importantly, we note that Lookbehind is the only method to outperform vanilla SAM and ASAM on ImageNet. We note, however, that the improvement of the loss-sharpness trade-off achieved by Lookbehind leads to a myriad of benefits on top of increased generalization performance, as demonstrated next.

Table 1: Generalization performance (validation accuracy %) of the different methods on several models trained on CIFAR-10, CIFAR-100, and ImageNet.

| Dataset | CIFAR-10 | | CIFAR-100 | | ImageNet |
|---|---|---|---|---|---|
| Model | ResNet-34 | WRN-28-2 | ResNet-50 | WRN-28-10 | ResNet-18 |
| SGD | $95.84_{\pm.13}$ | $93.58_{\pm.11}$ | $74.35_{\pm1.23}$ | $78.80_{\pm.08}$ | $69.91_{\pm.04}$ |
| Lookahead + SGD | $95.59_{\pm.21}$ | $94.01_{\pm.02}$ | $75.96_{\pm.12}$ | $78.53_{\pm.18}$ | $69.63_{\pm.12}$ |
| SAM | $95.80_{\pm.07}$ | $93.93_{\pm.20}$ | $76.57_{\pm.59}$ | $80.50_{\pm.06}$ | $70.01_{\pm.06}$ |
| Multistep-SAM | $95.72_{\pm.15}$ | $94.39_{\pm.09}$ | $77.03_{\pm.65}$ | $80.55_{\pm.06}$ | $69.92_{\pm.07}$ |
| Lookahead + SAM | $95.80_{\pm.11}$ | $93.97_{\pm.17}$ | $76.16_{\pm.98}$ | $80.09_{\pm.10}$ | $69.99_{\pm.07}$ |
| **Lookbehind + SAM** | $\mathbf{96.27_{\pm.07}}$ | $\mathbf{94.81_{\pm.22}}$ | $\mathbf{78.62_{\pm.48}}$ | $\mathbf{80.99_{\pm.02}}$ | $\mathbf{70.16_{\pm.08}}$ |
| ASAM | $96.32_{\pm.02}$ | $94.41_{\pm.09}$ | $78.62_{\pm.67}$ | $81.67_{\pm.28}$ | $70.15_{\pm.06}$ |
| Multistep-ASAM | $95.91_{\pm.14}$ | $95.06_{\pm.15}$ | $77.81_{\pm.52}$ | $81.67_{\pm.06}$ | $70.06_{\pm.01}$ |
| Lookahead + ASAM | $96.01_{\pm.15}$ | $94.28_{\pm.04}$ | $77.55_{\pm1.10}$ | $80.97_{\pm.17}$ | $70.00_{\pm.11}$ |
| **Lookbehind + ASAM** | $\mathbf{96.54_{\pm.21}}$ | $\mathbf{95.23_{\pm.01}}$ | $\mathbf{78.86_{\pm.29}}$ | $\mathbf{82.16_{\pm.09}}$ | $\mathbf{70.23_{\pm.22}}$ |

### 4.3 SHARPNESS ACROSS LARGE NEIGHBORHOOD REGIONS

We move on to analyzing the sharpness of the minima found at the end of training for each method. To do this, we measure the sharpness of the trained models using $m$-sharpness (Foret et al., 2021) by computing

$$\frac{1}{n} \sum_{M \in D} \max_{\|\epsilon\|_2 \leq r} \frac{1}{m} \sum_{s \in M} L_s(\phi + \epsilon) - L_s(\phi) \tag{4}$$

and

$$\frac{1}{n} \sum_{M \in D} \max_{\|\epsilon/|\phi|\|_2 \leq r} \frac{1}{m} \sum_{s \in M} L_s(\phi + \epsilon) - L_s(\phi) \tag{5}$$

for SAM and ASAM, respectively, where $D$ represents the training dataset, which is composed of $n$ minibatches $M$ of size $m$. To avoid ambiguity, we denote the radius used by $m$-sharpness as $r$. Instead of only measuring sharpness in close vicinity to the found solutions, *i.e.* using $r = 0.05$ as in Figure 1, we vary the radius $r$ over which $m$-sharpness is calculated. Particularly, we iterate over $r \in \{0.05, 0.5, 1.0, \dots, 5.0\}$ for SAM and $r \in \{0.5, 1.0, \dots, 5.0\}$ for ASAM.

The sharpness over different radii of the different methods, when also trained with different $\rho$, are shown in Figure 3. We observe that on top of Lookbehind improving sharpness at the nearby neighborhoods (as previously shown in Figure 1), SAM and ASAM models trained with Lookbehind also converge to flatter minima at the end of training, as measured on an extensive range of tested radii. This is consistent across training with different $\rho$ on both SAM and ASAM. Even though the minima found by the Lookahead and Multistep variants tend to have low sharpness when training with the default $\rho$, such benefits diminish at higher $\rho$.

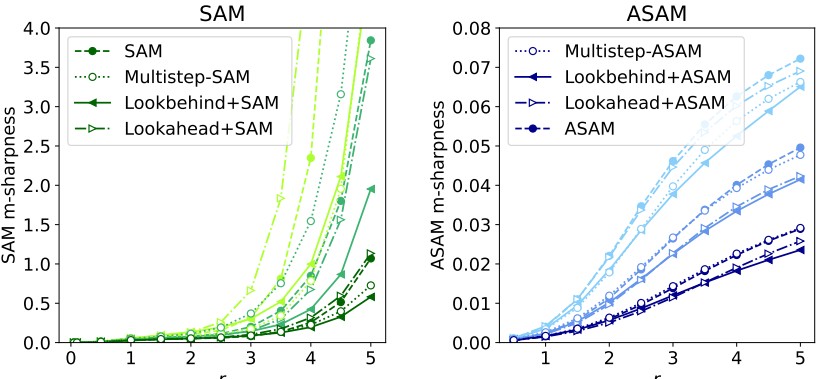

Figure 3: Sharpness at multiple $m$-sharpness's radius $r$ using ResNet-34 trained on CIFAR-10. Darker shades indicate training with higher neighborhood sizes $\rho$, ranging from $\rho \in \{0.05, 0.1, 0.2\}$ for SAM and $\rho \in \{0.5, 1.0, 2.0\}$ for ASAM. Lower sharpness is better.

### 4.4 MODEL ROBUSTNESS

We now assess model robustness against noisy weights. This is a particularly important use case when deployment models in highly energy-efficient hardware implementations that are prone to variabilities and noise (Xu et al., 2013; Kern et al., 2022; Spoon et al., 2021). Similar to previous works (Joshi et al., 2020; Mordido et al., 2022), we apply a multiplicative Gaussian noise to the model parameters $\phi$ after training in the form of $\phi \times \delta$, with $\delta \sim \mathcal{N}(1, \sigma^2)$ and update the batch normalization statistics after the noise perturbations. Robustness results are presented in Figure 4.

We see that Lookbehind shows the highest robustness observed by preserving the most amount of validation accuracy across the tested noise levels. This is observed for both SAM and ASAM on all models and datasets. We note that the benefits of using sharpness-aware minimization methods to increase model robustness to noisy weights were shown by previous works (Mordido et al., 2022). Our results share these findings and further show that Lookbehind considerably boosts the robustness benefits of training with SAM and ASAM across several models and datasets.

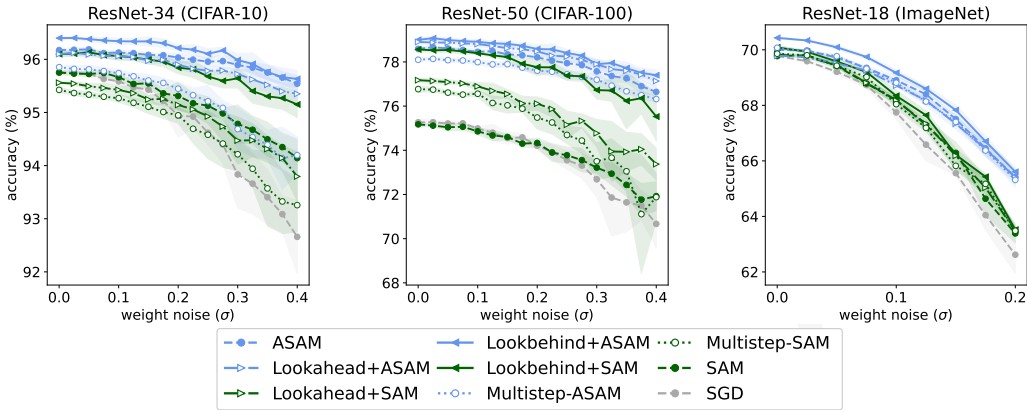

Figure 4: Robustness against noisy weights at inference time. We plot the mean and standard deviation over 10 and 3 inference runs for CIFAR-10/100 and ImageNet, respectively.

## 4.5 LIFELONG LEARNING

Lastly, we evaluate the methods in lifelong learning where a model with a limited capacity is trained on a stream of tasks. The goal is then to maximize performance across tasks without having access to previous data. In our experiments, we replicate the same setup used in Lookahead-MAML (Gupta et al., 2020), which is a lifelong learning method that combines the concept of slow and fast weights of Lookahead with meta-learning principles (Finn et al., 2017). Moreover, we replace Lookahead with Lookbehind, creating a novel algorithm called Lookbehind-MAML. Since meta-learning is out of the scope of this work, we implemented only the constant learning rate setting for simplicity, *i.e.* the C-MAML variant (Gupta et al., 2020).

We train a 3- and a 4-layer convolutional network on Split-CIFAR100 and Split-TinyImageNet, respectively. We report the following metrics by evaluating the model on the held-out data set: average accuracy (higher is better) and forgetting (lower is better). Additional details about the algorithms, training, and datasets are provided in Appendix A.5. The results are presented in Table 2. In the first setting, we do not use ER and directly compare our method with SGD, SAM, and Multistep-SAM. We observe that Lookbehind achieves the best performance both in terms of average accuracy and forgetting. In the second setting, we apply ER to the previous methods. Once again, we see an improvement when using our variant. Finally, we directly compare Lookahead-C-MAML with Lookbehind-C-MAML and also notice an overall performance improvement.

Table 2: Lifelong learning performance in terms of average accuracy (higher is better) and forgetting (lower is better) on Split-CIFAR100 and Split-TinyImageNet.

| Dataset | Split-CIFAR100 | | Split-TinyImagenet | |
|---|---|---|---|---|
| Metric | Avg. accuracy ↑ | Forgetting ↓ | Avg. accuracy ↑ | Forgetting ↓ |
| SGD | $58.41_{\pm4.95}$ | $22.74_{\pm4.85}$ | $43.48_{\pm0.80}$ | $26.51_{\pm0.71}$ |
| SAM | $57.81_{\pm1.05}$ | $23.27_{\pm0.57}$ | $56.34_{\pm1.72}$ | $20.39_{\pm1.83}$ |
| Multistep-SAM | $59.58_{\pm0.34}$ | $15.09_{\pm0.48}$ | $56.09_{\pm1.17}$ | $20.70_{\pm1.05}$ |
| **Lookbehind + SAM** | $\mathbf{59.93_{\pm1.54}}$ | $\mathbf{14.10_{\pm0.98}}$ | $\mathbf{56.60_{\pm0.68}}$ | $\mathbf{18.99_{\pm0.62}}$ |
| ER + SGD | $64.84_{\pm1.29}$ | $12.96_{\pm0.23}$ | $49.19_{\pm0.93}$ | $19.06_{\pm0.26}$ |
| ER + SAM | $68.28_{\pm1.30}$ | $13.98_{\pm0.42}$ | $65.59_{\pm0.19}$ | $9.89_{\pm0.14}$ |
| ER + Multistep-SAM | $65.49_{\pm4.10}$ | $15.20_{\pm2.53}$ | $65.75_{\pm0.16}$ | $9.90_{\pm0.09}$ |
| **ER + Lookbehind + SAM** | $\mathbf{68.87_{\pm0.79}}$ | $\mathbf{12.37_{\pm0.11}}$ | $\mathbf{65.91_{\pm0.27}}$ | $\mathbf{9.11_{\pm0.63}}$ |
| Lookahead-C-MAML | $65.44_{\pm0.99}$ | $13.96_{\pm0.86}$ | $61.93_{\pm1.55}$ | $11.53_{\pm1.11}$ |
| **Lookbehind-C-MAML** | $\mathbf{67.15_{\pm0.74}}$ | $\mathbf{12.40_{\pm0.49}}$ | $\mathbf{62.16_{\pm0.86}}$ | $\mathbf{11.21_{\pm0.44}}$ |

## 5 SENSITIVITY ANALYSIS

In this section, we analyze the sensitivity of Lookbehind to different hyper-parameter settings in terms of generalization performance (Sections 5.1, 5.2, and 5.3). For the following experiments, we used ResNet-34 and ResNet-50 models trained from scratch on CIFAR-10 and CIFAR-100, respectively. Training and hyperparameter search details are provided in Appendices A.3 and A.4.

### 5.1 SENSITIVITY TO THE INNER STEP $k$

Validation accuracies of the different methods when using different $k$ are presented in Figure 5. We observe that Lookbehind is the only method that consistently outperforms the SAM and ASAM baselines on both CIFAR-10 and CIFAR-100, across all the tested inner steps $k$. Interestingly, our method tends to keep improving when increasing $k$, while this trend is not observed for either the Lookahead or the Multistep variants. Moreover, we see that Multistep-SAM/ASAM does not provide a clear improvement over the respective SAM and ASAM baselines, as previously discussed in prior work (Foret et al., 2021; Andriushchenko & Flammarion, 2022). On the other hand, the Lookahead variants show a slight improvement over Multistep, particularly when combining Lookahead with SAM and ASAM on CIFAR-10 and SAM on CIFAR-100. Overall, we see that Lookbehind reaches the highest validation accuracy on every tested model and dataset configuration when combined with both SAM and ASAM.

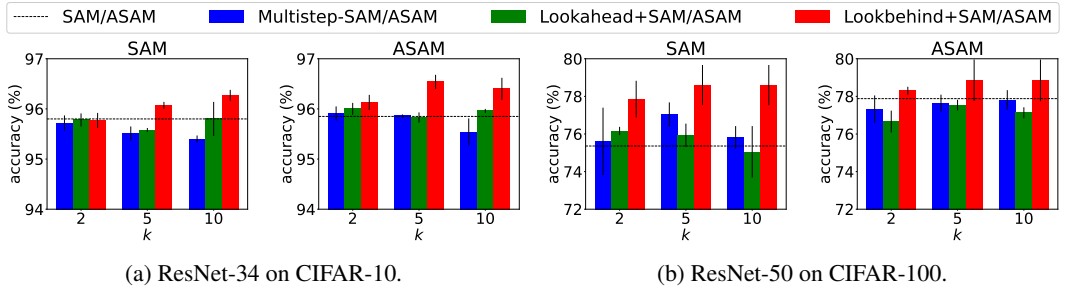

(a) ResNet-34 on CIFAR-10.  (b) ResNet-50 on CIFAR-100.

Figure 5: Comparison of generalization performance (validation accuracy %) between Multistep-SAM/SAM, Lookahead + SAM/ASAM, and Lookbehind + SAM/ASAM. The vanilla SAM and ASAM baselines with default $\rho$ are represented by the horizontal, dotted line.

### 5.2 SENSITIVITY TO THE OUTER STEP SIZE $\alpha$

The validation accuracies of Lookbehind across different $\alpha$ and $k$ are presented in Figure 6. We see that Lookbehind always improves over the baselines when considering the full grid search. This is also reflected in a finer-grained manner, where Lookbehind improves over the baselines in all $k$, except $k = 2$ on SAM and CIFAR-10. We notice a diagonal trend, suggesting there is a relation between $\alpha$ and $k$. Specifically, the results suggest that a higher $\alpha$ is better when increasing $k$. These results show that Lookbehind is robust to the choice of $k$ and $\alpha$ and while tuning these hyperparameters may improve performance, using a default high $\alpha$ (*e.g.* 0.5 or 0.8) with high $k$ (*e.g.* 5 or 10) often results in good performance.

### 5.3 SENSITIVITY TO THE NEIGHBORHOOD SIZE $\rho$

We now analyze the effects of training with increasing $\rho$ with the different methods. Results are presented in Figure 7. We see that our method is the only one that consistently outperforms SAM and ASAM across all the tested $\rho$. As previously suggested, significantly increasing $\rho$ in the SAM and ASAM baselines, *e.g.* $\rho = 0.5$ and $\rho = 5.0$, respectively, decreases performance relative to their default $\rho$, *e.g.* $\rho = 0.05$ and $\rho = 0.5$, respectively. Notwithstanding, we note that ASAM shows higher relative robustness to higher $\rho$ than SAM, indicated by ASAM's ability to continue increasing performance on up to $4\times$ the default neighborhood size, *i.e.* from $\rho = 0.5$ to $\rho = 2.0$. Lastly, we note that the Lookbehind and Multistep variants show similar trends as the SAM and

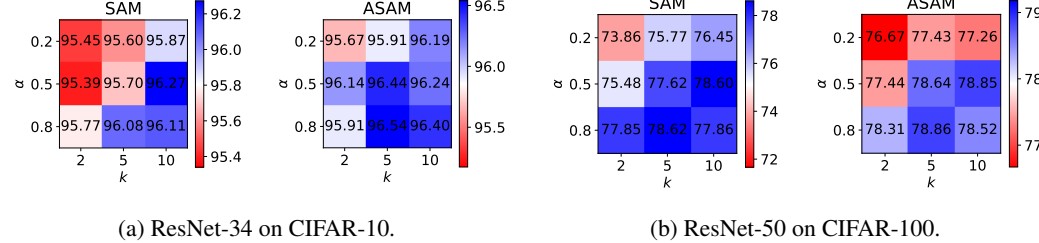

(a) ResNet-34 on CIFAR-10.  (b) ResNet-50 on CIFAR-100.

Figure 6: Sensitivity of Lookbehind to $\alpha$ and $k$ when combined with SAM and ASAM in terms of generalization performance (validation accuracy %). The validation accuracies of the SAM and ASAM variants are presented in the middle of the heatmap (white middle point). All models were trained with the default $\rho$. Blue represents an improvement in terms of validation accuracy over such baselines, while red indicates a degradation in performance.

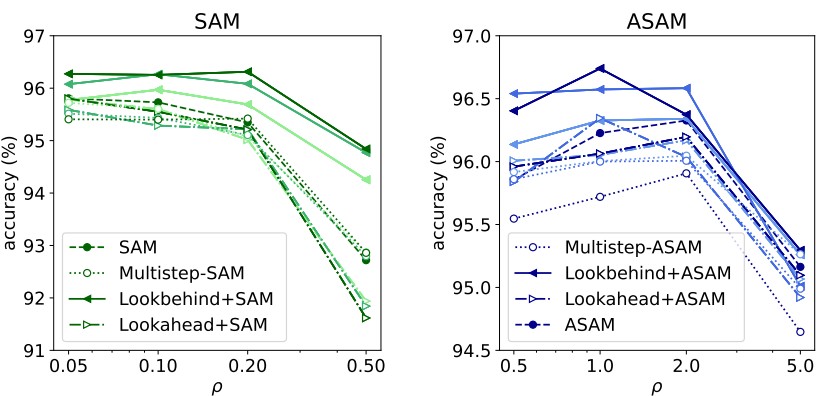

Figure 7: Validation accuracies with different trained $\rho$ for the different methods using ResNet-34 trained on CIFAR-10. Darker shades represent larger inner steps $k$, ranging from $k \in \{2, 5, 10\}$.

ASAM baselines. Overall, we observe that Lookbehind is more robust to the choice of $\rho$ compared to the other methods.

## 6 ADAPTIVE $\alpha$

Lookbehind adds two additional hyperparameters to SAM/ASAM – just as the Lookahead optimizer adds two hyperparameters to SGD - which introduces additional hyperparameter tuning on top of $\rho$ and $\eta$. To mitigate this added complexity in settings where computational resources are scarce, we investigate if we can remove the need to tune $\alpha$ by instead computing it analytically during training. We refer to this adaptive formulation of $\alpha$ as $\alpha^*$. The main idea is to set $\alpha^*$ proportionally to the alignment of the gradients obtained during the multiple ascent steps:

$$\alpha^* = (\cos(\theta) + 1)/2\,, \tag{6}$$

where $\theta$ is defined by the angle between the first gathered gradient and the final update direction:

$$\theta = \frac{(\phi_{t,1} - \phi_t) \cdot (\phi_{t,k} - \phi_t)}{\|\phi_{t,1} - \phi_t\|_2 \cdot \|\phi_{t,k} - \phi_t\|_2}\,. \tag{7}$$

If the gradients are completely aligned, then $\alpha^* = 1$. On the other hand, if the gradients are not aligned, then $0 \leq \alpha^* < 1$, with lower values representing lower alignment.

Results when using Lookbehind with a static $\alpha$ and a dynamic $\alpha^*$ are presented in Table 3. Overall, we observe that using an adaptive $\alpha$ is a viable alternative to tuning a static $\alpha$ in instances where compute is scarce. Note that our goal with adaptive $\alpha$ is not necessarily to outperform static $\alpha$

Table 3: Generalization performance (validation acc. %) of Lookbehind with static and adaptive $\alpha$.

| Dataset | CIFAR-10 | | CIFAR-100 | | ImageNet |
|---|---|---|---|---|---|
| Model | ResNet-34 | WRN-28-2 | ResNet-50 | WRN-28-10 | ResNet-18 |
| Lookbehind + SAM | $96.27_{\pm.07}$ | $94.81_{\pm.22}$ | $\mathbf{78.62_{\pm.48}}$ | $\mathbf{80.99_{\pm.02}}$ | $\mathbf{70.16_{\pm.08}}$ |
| + adaptive $\alpha$ | $\mathbf{96.33_{\pm.04}}$ | $\mathbf{94.88_{\pm.12}}$ | $78.33_{\pm.36}$ | $80.86_{\pm.13}$ | $70.07_{\pm.12}$ |
| Lookbehind + ASAM | $96.54_{\pm.21}$ | $\mathbf{95.23_{\pm.01}}$ | $78.86_{\pm.29}$ | $\mathbf{82.16_{\pm.09}}$ | $\mathbf{70.23_{\pm.22}}$ |
| + adaptive $\alpha$ | $\mathbf{96.57_{\pm.03}}$ | $95.08_{\pm.15}$ | $\mathbf{78.89_{\pm.45}}$ | $81.86_{\pm.22}$ | $70.16_{\pm.08}$ |

but instead to achieve competitive performance while having one less hyperparameter. Importantly, we emphasize that Lookbehind with adaptive $\alpha$ consistently outperforms all the compared methods presented in Table 1, similarly to static $\alpha$. Due to space constraints, we refer the reader to Appendix A.1.4 for additional analysis on how $\alpha^*$ varies during training. Moreover, additional discussions are provided in Appendix A.1.

## 7 RELATED WORK

Sharpness-aware minimization (SAM) (Foret et al., 2021) is an attempt to improve generalization by finding solutions with both low loss value and low loss sharpness. This is achieved by minimizing an estimation of the maximum loss over a neighborhood region around the parameters. There is currently a lot of active work that focuses on improving SAM. More specifically, modifications of the original SAM algorithm were proposed to further improve generalization performance (Zhuang et al., 2022; Kim et al., 2022; Kwon et al., 2021; Liu et al., 2022b) and efficiency (Du et al., 2022c; Zhou et al., 2022; Liu et al., 2022a). Performing multiple ascent steps was present in Foret et al. (2021), however, the improvements over single ascent step SAM were either insignificant or even shown to degrade performance in some settings (Andriushchenko & Flammarion, 2022).

SAM's benefits have transcended improving generalization performance, ranging from higher robustness to label noise (Foret et al., 2021; Kwon et al., 2021), lower quantization error (Liu et al., 2021b), and less sensitivity to data imbalance (Liu et al., 2021a). Here, on top of analyzing the benefits of Lookbehind on generalization performance, we focused on further improving the recently observed benefits of SAM on improving robustness against noisy weights (Kim et al., 2022; Mordido et al., 2022) and reducing catastrophic forgetting in lifelong learning (Mehta et al., 2021).

Closest to our work, Kim et al. (2023) concurrently conducted a similar study by averaging the gradients obtained during multiple SAM ascent steps. One of the differences is the decoupling of the inner step $k$ and the outer step size $\alpha$ in our approach, which allows us to seek optimal combinations between these two hyperparameters. In fact, as depicted in Figures 6 and 13, $\alpha = 1/k$ is generally not the best overall $\alpha$ to use, including when determining $\alpha^*$ (Figure 10). We also extend the empirical discussions by applying our method with ASAM, which often produces superior results (as shown in Table 1). Additionally, we explore the applicability of our approach to lifelong learning (by applying our method with MAML) and robustness settings.

## 8 CONCLUSION

In this work, we proposed the Lookbehind optimizer, which can be plugged on top of existing sharpness-aware training methods to improve performance over a variety of benchmarks. Our experiments show that our method improves the generalization performance on multiple models and datasets, increases model robustness, and promotes the ability to continuously learn in lifelong learning settings. Even though the goal of this work is to tackle the lack of performance due to a poor sharpness-loss trade-off, another important issue inherent to any multiple ascent step SAM method is the computational overhead which increases training time by a factor $k$. In the future, it would be interesting to investigate how to improve the efficiency of multiple ascent steps, *e.g.* by switching the minibatch at each inner step of Lookbehind.

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

# A    APPENDIX

Here, we provide additional discussions (Section A.1) and more information on the Lookahead+SAM baseline (Section A.2). Moreover, we present further details on the training procedures (Section A.3) and the lifelong learning setup (Section A.5). We also provide additional sensitivity analysis across all tested models (Section A.6). Lastly, we present further comparisons with additional methods (Section A.7) and training setups (Section A.8).

## A.1    ADDITIONAL DISCUSSIONS

In this Section, we analyze additional limitations of our work (Section A.1.1) as well as additional studies to better understand the behavior of Lookbehind. In particular, we showcase the benefits of Lookbehind at different training stages (Section A.1.2), the advantage of going farther away from the original solution as performing multiple ascent steps instead of staying within a neighborhood size $\rho$ (Section A.1.3), and how the values of $\alpha^*$ evolve during training (Section A.1.4).

### A.1.1    LIMITATIONS

One drawback of our approach is the introduction of two new hyperparameters to SAM/ASAM. This was partially addressed in Section 6 by removing the need to fine-tune $\alpha$. Nevertheless, even with the adaptive $\alpha$ variant, our method still introduces one more hyperparameter. Since tuning hyperparameters requires more compute, the comparison with baselines with less hyperparameters is only reasonable to the extent that the baselines are not subject to computational constraints that might limit their performance, *e.g.* by not training for long enough. However, we argue that this was not the case in our experimental setup, and additional training would be unlikely to improve the performances reported for the SAM/ASAM baselines. To corroborate this, we show the average number of epochs at which the best SAM and ASAM baseline configurations achieved the best validation accuracy in Table 4. We observe that the best-performing model checkpoints were not completed at the very end of training (*e.g.* last epoch) across our experimental setup, suggesting there was prior performance saturation before training finished.

Table 4: Average number of epochs at which the SAM and ASAM baselines achieved the best validation accuracy across the different models and datasets. The models were trained for a total of 200 epochs for CIFAR-10/100 and 90 epochs for ImageNet.

| Dataset | CIFAR-10 | | CIFAR-100 | | ImageNet |
| Model | ResNet-34 | WRN-28-2 | ResNet-50 | WRN-28-10 | ResNet-18 |
|---|---|---|---|---|---|
| SAM | $164.66_{\pm 9.87}$ | $141.33_{\pm 15.45}$ | $167.66_{\pm 21.06}$ | $172.00_{\pm 9.00}$ | $83.66_{\pm 2.05}$ |
| ASAM | $164.00_{\pm 12.83}$ | $158.66_{\pm 29.45}$ | $178.66_{\pm 3.77}$ | $179.50_{\pm 4.50}$ | $86.00_{\pm 2.16}$ |

### A.1.2    BENEFITS OF LOOKBEHIND AT DIFFERENT STAGES DURING TRAINING

SAM has been shown to find better generalizable minima within the same basin as SGD. In other words, SAM's implicit bias mostly improves the generalization of SGD when switching from SGD to SAM toward the end of training (Andriushchenko & Flammarion, 2022). Interestingly, the aforementioned results also suggest that SAM and SGD do not guide optimization toward different basins from early on in training. Here, we conduct a similar study by analyzing how switching from SAM/ASAM to Lookbehind+SAM/ASAM, and vice-versa, impacts generalization performance at different stages during training.

The generalization performances of starting training with SAM/ASAM and switching to Lookbehind at different training stages are shown in Figure 8a. We observe that Lookbehind's benefits are mostly achieved early on in training, suggesting that Lookbehind guides the optimization to converge to a different basin of the loss landscape than SAM. Such findings are confirmed by also switching from Lookbehind to SAM/ASAM (Figure 8b).

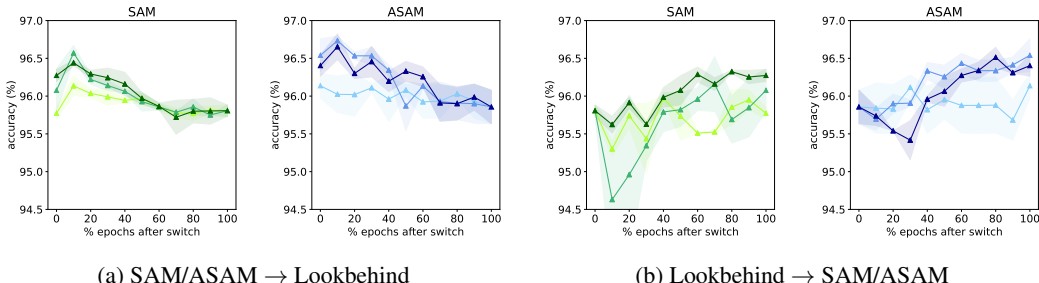

(a) SAM/ASAM → Lookbehind                    (b) Lookbehind → SAM/ASAM

Figure 8: Impact of switching from SAM/ASAM to Lookbehind + SAM/ASAM (a), and vice-versa (b), at different epochs throughout training in terms of validation accuracy using ResNet-34 trained on CIFAR-10. Darker shades represent larger inner steps $k$, ranging from $k \in \{2, 5, 10\}$. For Lookbehind, we pick the best $\alpha$ configuration for each $k \in \{2, 5, 10\}$ using the default $\rho$, which is also used for the SAM/ASAM baselines.

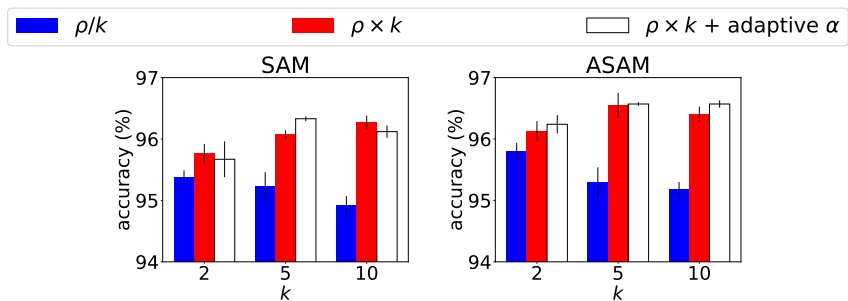

Figure 9: Comparison of generalization performance (validation accuracy %) on ResNet-34 trained on CIFAR-10 between staying up to a neighborhood $\rho$ or $\rho \times k$. We also plot the performance of adaptive $\alpha$ in the latter setting.

### A.1.3 STAYING WITHIN A NEIGHBORHOOD SIZE $\rho$ OR $\rho/k$

As a wrapper to SAM methods, Lookbehind's practicality is enhanced when there is no need to re-tune the default $\rho$ of the sharpness-aware minimizer. To study this, we used the default $\rho$ suggested by SAM and ASAM and investigated if staying within a neighborhood $\rho$ of the original solution is more advantageous than increasing the neighborhood up to $\rho \times k$, as presented so far throughout our paper. For this new variant, we reduce the neighborhood size to $\rho/k$ as the step size for each ascent step. Hence, after $k$ ascent steps we will be at a maximum distance $\rho$ from the original point if all gradients align. We also remove linear interpolation and simply set the descent step size to $\eta$. Results using the default $\rho$ for SAM and ASAM are presented in Figure 9.

We observe that going farther away as we perform the ascent steps consistently outperforms staying within a neighborhood $\rho$ of the original solution. In other words, $\rho \times k$ is better than $\rho/k$ when using the default $\rho$ of SAM and ASAM. This is a convenient insight since we show that tuning the hyperparameter $\rho$ is not necessary when using the former setting. Moreover, this also allows us to learn $\alpha$ dynamically, which is shown to enhance performances in some settings. This suggests that it is beneficial to not only "look behind" within a neighborhood of $\rho \times k$, but also that taking a dynamic descent step size to perform the final update based on the alignment of the aggregated gradients is an effective way of enhancing performance across different $k$.

### A.1.4 CHANGE OF $\alpha^*$ DURING TRAINING

We show how adaptive $\alpha$ changes throughout training in Figure 10. We notice an expected trend based on the values of $k$, with higher $k$ leading to lower $\alpha^*$ due to less gradient alignment. Even though $\alpha$ is independent of the inner step learning rate $\eta$, we are decreasing $\eta$ by a factor of 10 every 50 epochs in our training setup, which leads to drastic changes in model performance and loss

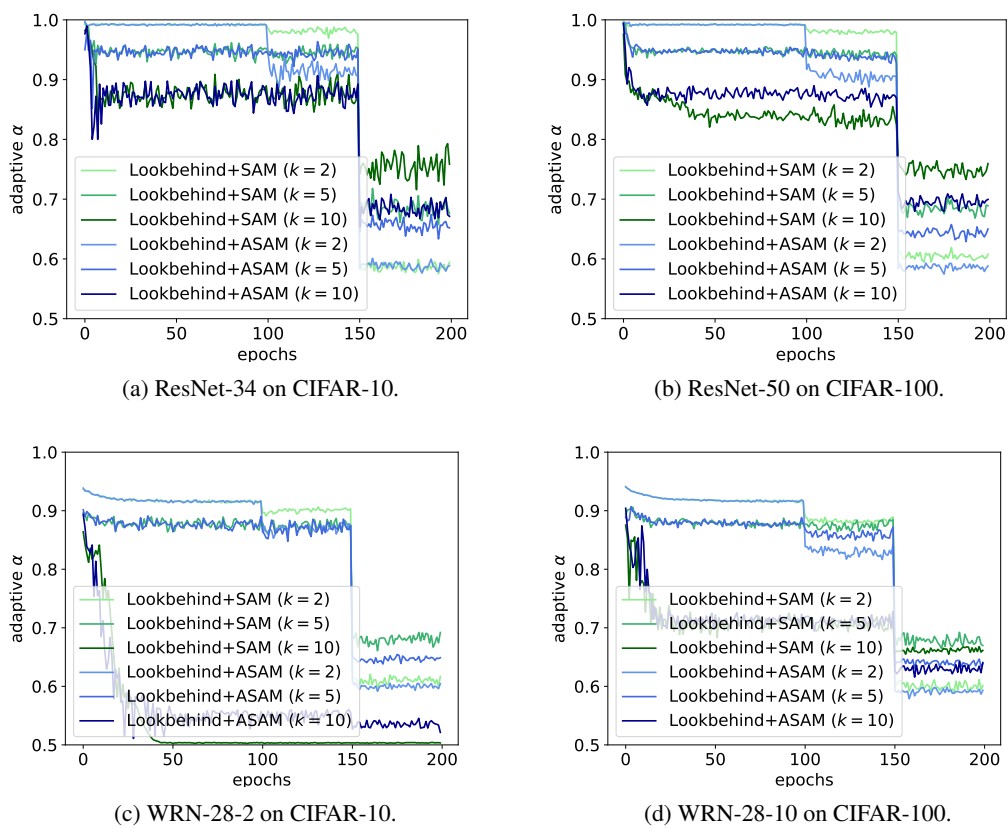

Figure 10: Analysis of how adaptive $\alpha$ evolves during training.

---

**Algorithm 2** Lookahead+SAM

**Require:** Initial parameters $\phi_0$, loss function $L$, inner steps $k$, slow weights step size $\alpha$, fast weights step size $\eta$, neighborhood size $\rho$, training set $D$

1: **for** $t = 1, 2, \ldots$ **do**
2:     $\phi_{t,0} \leftarrow \phi_{t-1}$
3:     **for** $i = 1, 2, \ldots, k$ **do**
4:         Sample mini-batch $d \sim D$
5:         $\epsilon \leftarrow \rho \dfrac{\nabla L_d(\phi_{t,i-1})}{\|\nabla L_d(\phi_{t,i-1})\|_2}$
6:         $\phi_{t,i} \leftarrow \phi_{t,i-1} - \eta \nabla_{L_d}(\phi_{t,i-1} + \epsilon)$
7:     **end for**
8:     $\phi_t \leftarrow \phi_{t-1} + \alpha(\phi_{t,k} - \phi_{t-1})$
9: **end for**
10: **return** $\phi$

Figure 11: Combination of Lookahead with SAM.

landscape. This in turn seems to lead to an increase in the misalignment of the aggregated gradients which decreases the adaptive $\alpha$ values later on in training.

## A.2 Lookahead+SAM

Lookahead (Zhang et al., 2019) was introduced to reduce variance during training, with the end goal of improving performance and robustness to hyper-parameter settings. Given an optimizer, Lookahead uses slow and fast weights to improve its training stability. The algorithm "looks ahead" by updating the fast weights $k$ times in an inner loop, while the slow weights are updated by performing a linear interpolation to the final fast weights (after the inner loop ends). In our analysis and experiments, we use Lookahead with sharpness-aware methods by applying single-step SAM and ASAM as the inner optimizers. The main goal of these baselines is to use Lookahead to stabilize sharpness-aware optimizers when training with large $\rho$. An illustration of Lookahead+SAM is presented in Figure 11 (right).

Similarly to our method, Lookahead+SAM uses slow weights ($\phi_t$, $\phi_{t+1}$, $\cdots$) and fast weights ($\phi_{t,1}$, $\cdots$, $\phi_{t,k}$). However, the slow weights are updated after each SAM update (composed of a single ascent and descent step), while the slow weights are updated toward the fast weights through linear interpolation after $k$ steps ($\phi_{t+1}$). In contrast, Lookbehind+SAM's fast and slow weights are obtained during a given iteration. In particular, while the fast weights are updated as we "look behind", the slow weights are updated after $k$ ascent steps are performed (c.f. Figure 2).

The pseudo-code for combining Lookahead with SAM is presented in Figure 11 (left). Just like Lookahead, Lookahead+SAM maintains a set of slow weights and fast weights, which are synchronized at the beginning of every outer step (line 2). Then, the fast weights are updated $k$ times (looking forward) using a standard SAM update with a single ascent (line 5) and descent step (line 6). After $k$ such SAM steps, the slow weights are updated by linearly interpolating to the final fast weights (line 8) (1 step back). It is worth noting that a new minibatch is sampled at every inner step (line 4). Combining Lookahead with ASAM follows the same procedure, except using the component-wise rescaling (equation 3) in line 5.

Although Lookbehind+SAM and Lookahead+SAM share a similar nature, they exhibit notable distinctions. Firstly, in addition to synchronizing the fast weights, Lookbehind also synchronizes the perturbed fast weights. Furthermore, the minibatch is sampled before the inner loop. Moreover, at each inner step, Lookbehind performs $k$ ascent steps of SAM. The distinction between the two algorithms leads to divergent behavior in the training objective and is related to Lookahead+SAM and Lookbehind+SAM having different goals: while Lookahead+SAM aims at stabilizing single-step SAM with large neighborhood sizes $\rho$, Lookbehind aims to perform multiple ascent steps while maintaining a good balance between sharpness and training accuracy.

In other words, Lookbehind focuses on curbing the variance arising from gradients gathered during multiple ascent steps within a single iteration. In contrast, Lookahead+SAM targets variance stemming from sequential descent steps performed across iterations. Hence, our goal is to reduce the variance of looking behind, not ahead.

## A.3 Training details

For CIFAR-10/100, we trained each model for 200 epochs with a batch size of 128, starting with a learning rate of 0.1 and dividing it by 10 every 50 epochs. For ImageNet, we use 1000 classes and an image size of 224x224 and trained each model for 90 epochs with a batch size of 400, starting with a learning rate of 0.1 and dividing it by 10 every 30 epochs. All models were trained using SGD with momentum set to 0.9 and weight decay of 1e-4. We trained the CIFAR-10/100 models using one RTX8000 NVIDIA GPU and 1 CPU core, and the ImageNet models using one A100 GPU and 6 CPU cores. For CIFAR-10/100, we used the architecture implementations in `https://github.com/kuangliu/pytorch-cifar`. For ImageNet, we used the ResNet-18 implementation provided by PyTorch [1].

## A.4 Hyperparameter search

For Table 1, we only perform hyperparameter search for $\rho \in \{0.05, 0.1, 0.2\}$ for all vanilla SAM and $\rho \in \{0.5, 1.0, 2.0\}$ for all vanilla ASAM baselines, and report the validation results with the

---

[1] `https://pytorch.org/vision/main/models/generated/torchvision.models.resnet18.html`

best $\rho$. For the rest of the methods, we used the default $\rho$, *i.e.* as presented in the original SAM Foret et al. (2021) and ASAM Kwon et al. (2021) papers. Particularly we used $\rho$ of 0.05, 0.1, and 0.05 for SAM and 0.5, 1.0, and 1.0 for ASAM when training on CIFAR-10, CIFAR-100, and ImageNet, respectively. For CIFAR-10/100, we use $k \in \{2, 5, 10\}$ and $\alpha \in \{0.2, 0.5, 0.8\}$ (when applicable) for the multiple step methods. For ImageNet, we use $k = 2$ and $\alpha \in \{0.2, 0.5, 0.8\}$ (when applicable).

For Figure 3, we report the best $k$ and $\alpha$ configurations for all methods, *i.e.* with the lowest sharpness at the highest $r$.

For Figure 4, we report the most robust model using $k \in \{2, 5, 10\}$ and $\alpha \in \{0.2, 0.5, 0.8\}$ for CIFAR-10/100. For ImageNet, we use $k = 2$ and $\alpha \in \{0.2, 0.5, 0.8\}$. For the SAM and ASAM baselines, we pick the most robust $\rho \in \{0.05, 0.1, 0.2, 0.5\}$ and $\rho \in \{0.5, 1.0, 2.0, 5.0\}$, respectively.

For Figure 5 we report the default neighborhood sizes for the SAM ($\rho = 0.05$ and 0.1 for CIFAR-10 and CIFAR-100, respectively) and ASAM baselines ($\rho = 0.5$ and 1.0 for CIFAR-10 and CIFAR-100, respectively). We show the best hyper-parameter configuration over $k \in \{2, 5, 10\}$ and $\alpha \in \{0.2, 0.5, 0.8\}$ for Lookbehind and Lookahead, and $k \in \{2, 5, 10\}$ for Multistep.

For Figure 7, we report the best $\alpha$ configuration for Lookahead and Lookbehind.

## A.5 Lifelong Learning

We replicated the experimental setup from Lookahead-MAML (Gupta et al., 2020) and report the results for all baselines where the models were trained for 10 epochs per task. Additionally, we combined the different methods with episodic replay (ER) (Chaudhry et al., 2019), which maintains a memory of a subset of the data from each task and uses it as a replay buffer while training on new tasks. We test both settings (with and without ER) in our experiments. We used two datasets: Split-CIFAR100 and Split-TinyImageNet. The Split-CIFAR100 benchmark is designed by splitting the 100 classes in CIFAR-100 into 20 5-way classification tasks. Similarly, Split-TinyImageNet is designed by splitting 200 classes into 40 5-way classification tasks. In both cases, the task identities are provided to the model along with the dataset. Each model has multi-head outputs, *i.e.* each task has a separate classifier.

We provide the grid search details for finding the best set of hyper-parameters for both datasets and all baselines in Table 5. We train the model on the training set and report the best hyper-parameters based on the highest accuracy on the test set in Table 6. Here, we report the hyper-parameter set for each method (with or without ER) as follows:

- SGD: $\{\eta\}$
- SAM: $\{\eta, \rho\}$
- Multistep-SAM: $\{\eta, \rho, k\}$
- Lookbehind + SAM: $\{\eta, \rho, k, \alpha\}$
- Lookbehind-C-MAML: $\{\eta, \rho, k, \alpha\}$

We refer to Gupta et al. (2020) for the best hyper-parameters of Lookahead-C-MAML. We evaluated all models using the following metrics:

- **Average accuracy** (Lopez-Paz & Ranzato, 2017): the average performance of the model across all the previous tasks is defined by $\frac{1}{t} \sum_{\tau=1}^{t} a_{t,\tau}$, where $a_{t,\tau}$ is the accuracy on the test set of task $\tau$ when the current task is $t$.
- **Forgetting** (Chaudhry et al., 2018): the average forgetting that occurs after the model is trained on several tasks is computed by $\frac{1}{t-1} \sum_{\tau=1}^{t-1} \max_{t' \in \{1, \ldots, t-1\}} (a_{t',\tau} - a_{t,\tau})$, where $t$ represents the latest task.

We report the average accuracy and forgetting after the models were trained on all tasks for both datasets.

The pseudo-code for Lookahead-C-MAML and Lookbehind-C-MAML is presented in Figure 12.

---

**Algorithm 3** Lookahead-C-MAML (Gupta et al., 2020)

---

**Require:** Initial parameters $\phi_0^0$, inner loss function $\ell$, meta
     loss function $L$, step size $\eta$, training set $D_t$ of task $t$,
     number of epochs $E$

1: $j \leftarrow 0$
2: $R \leftarrow \{\}$
3: **for** $t = 1, 2, \ldots$ **do**
4:     Sample batch $d_t \sim D_t$
5:     **for** $e = 1, 2, \ldots, E$ **do**
6:        **for** mini-batch $b$ **in** $d_t$ **do**
7:           $k \leftarrow \text{sizeof}(b)$
8:           $b_m \leftarrow \text{Sample}(R) \cup b$
9:           **for** $k' = 0$ **to** $k - 1$ **do**
10:             Push $b[k']$ to R
11:             $\phi_{k'+1}^j \leftarrow \phi_{k'}^j - \eta \nabla_{\phi_{k'}^j} \ell_t(\phi_{k'}^j, b[k'])$
12:           **end for**
13:           $\phi_0^{j+1} \leftarrow \phi_0^j - \eta \nabla_{\phi_0^j} L_t(\phi_k^j, b_m)$
14:           $j \leftarrow j + 1$
15:        **end for**
16:     **end for**
17: **end for**
18: **return** $\phi$

---

---

**Algorithm 4** Lookbehind-C-MAML (ours)

---

**Require:** Initial parameters $\phi_{0,0}^0$, inner loss function $\ell$, meta loss function $L$, inner steps $k$, step
    size $\eta$, outer step size $\alpha$, neighborhood size $\rho$, training set $D_t$ of task $t$, number of epochs
    $E$

1: $j \leftarrow 0$
2: $R \leftarrow \{\}$
3: **for** $t = 1, 2, \ldots$ **do**
4:     $\phi_{t,0}^j \leftarrow \phi_{t-1,0}^j$
5:     Sample batch $d_t \sim D_t$
6:     **for** $e = 1, 2, \ldots, E$ **do**
7:        $\phi_{t,0}^{\prime j} \leftarrow \phi_{t,0}^j$
8:        **for** $k' = 0$ **to** $k - 1$ **do**
9:           Sample mini-batch $b \sim d_t$ of size $k$ without replacement
10:           $b_m \leftarrow \text{Sample}(R) \cup b$
11:           Push $b[k']$ to R
12:           $\epsilon \leftarrow \rho \dfrac{\nabla_{\ell_t}(\phi_{t,k'}^{\prime j}, b[k'])}{\|\nabla_{\ell_t}(\phi_{t,k'}^{\prime j}, b[k'])\|_2}$
13:           $\phi_{t,k'+1}^j \leftarrow \phi_{t,k'}^j - \eta \nabla_{\ell_t}(\phi_{t,k'}^{\prime j} + \epsilon, b[k'])$
14:        **end for**
15:        $\phi_{t,k}^j \leftarrow \phi_{t,0}^j + \alpha(\phi_{t,k}^j - \phi_{t,0}^j)$
16:        $\epsilon \leftarrow \rho \dfrac{\nabla_{\phi_{t,0}^j} L_t(\phi_{t,k}^j, b_m)}{\|\nabla_{\phi_{t,0}^j} L_t(\phi_{t,k}^j, b_m)\|_2}$
17:        $\phi_{t,0}^{j+1} \leftarrow \phi_{t,0}^j - \eta \nabla_{\phi_{t,0}^j} L_t(\phi_{t,0}^j + \epsilon, b_m)$
18:        $j \leftarrow j + 1$
19:     **end for**
20: **end for**
21: **return** $\phi$

---

Figure 12: Implementations of Lookahead-C-MAML (top) and Lookbehind-C-MAML (bottom).

Table 5: Details on the hyper-parameter grid search used for the lifelong learning experiments.

| Hyper-parameters | Values |
|---|---|
| step size ($\eta$) | $\{0.3, 0.1, 0.03, 0.01, 0.003, 0.001, 0.0003, 0.0001, 0.00003, 0.00001\}$ |
| inner steps ($k$) | $\{2, 5, 10\}$ |
| outer step size ($\alpha$) | $\{0.1, 0.2, 0.5, 0.8, 1.0\}$ |
| neighborhood size ($\rho$) | $\{0.005, 0.01, 0.05, 0.1\}$ |

Table 6: Best hyper-parameter settings for the different lifelong learning methods.

| Methods | Split-CIFAR100 | Split-TinyImagenet |
|---|---|---|
| SGD | $\{0.03\}$ | $\{0.03\}$ |
| SAM | $\{0.03, 0.05\}$ | $\{0.03, 0.05\}$ |
| Multistep-SAM | $\{0.01, 0.01, 2\}$ | $\{0.03, 0.05, 2\}$ |
| Lookbehind + SAM | $\{0.1, 0.05, 10, 0.1\}$ | $\{0.01, 0.05, 10, 0.1\}$ |
| ER + SAM | $\{0.1, 0.05\}$ | $\{0.03, 0.1\}$ |
| ER + Multistep-SAM | $\{0.1, 0.05, 10\}$ | $\{0.03, 0.1, 10\}$ |
| ER + Lookbehind + SAM | $\{0.03, 0.05, 10, 0.2\}$ | $\{0.01, 0.1, 5, 0.5\}$ |
| Lookbehind-C-MAML | $\{0.03, 0.005, 2, 1\}$ | $\{0.03, 0.1, 2, 1\}$ |

## A.6    SENSITIVITY TO $\alpha$ AND $k$

We measure the sensitivity to $\alpha$ and $k$ of Lookbehind and Lookahead on additional models in Figures 13 and 14, respectively. Similarly to the sensitivity results presented in the main paper, we observe that Lookbehind is more robust to the choice of $\alpha$ and $k$ and is able to improve on the SAM and ASAM baselines more significantly and consistently than Lookahead.

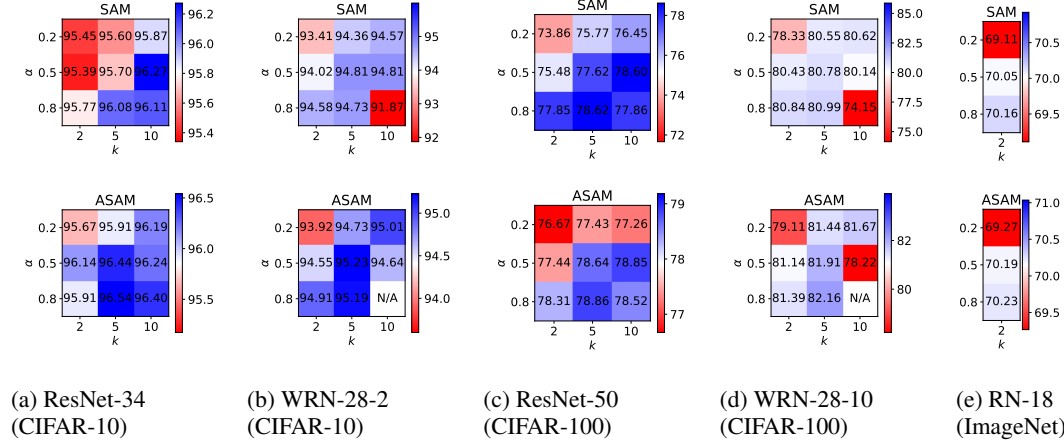

(a) ResNet-34
(CIFAR-10)

(b) WRN-28-2
(CIFAR-10)

(c) ResNet-50
(CIFAR-100)

(d) WRN-28-10
(CIFAR-100)

(e) RN-18
(ImageNet)

Figure 13: Sensitivity of Lookbehind to $\alpha$ and $k$ when combined with SAM and ASAM in terms of generalization performance (validation accuracy %). The validation accuracies of the SAM and ASAM variants are presented in the middle of the heatmap (white middle point). All models were trained with the default $\rho$. Blue represents an improvement in terms of validation accuracy over such baselines, while red indicates a degradation in performance. Experiments represented as "N/A" indicate instances where at least one seed failed to converge.

## A.7    ADDITIONAL METHOD COMPARISONS

We compare Lookbehind with the Multistep-SAM/ASAM with gradient averaging (Kim et al., 2023) in Table 7. We observe that both our method variants, *i.e.* Lookbehind with and without an adaptive $\alpha$, consistently outperform the additional baseline on all tested models and datasets.

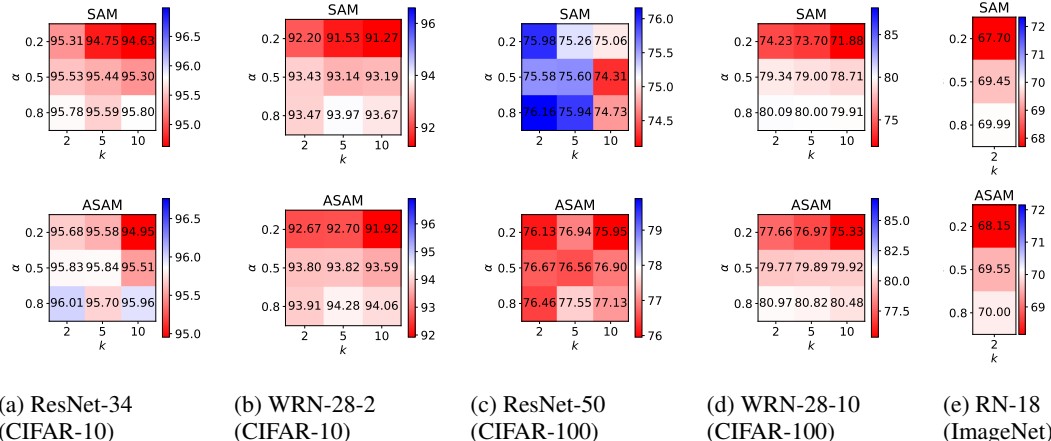

(a) ResNet-34
(CIFAR-10)

(b) WRN-28-2
(CIFAR-10)

(c) ResNet-50
(CIFAR-100)

(d) WRN-28-10
(CIFAR-100)

(e) RN-18
(ImageNet)

Figure 14: Sensitivity of Lookahead to $\alpha$ and $k$ when combined with SAM and ASAM in terms of generalization performance (validation accuracy %). The validation accuracies of the SAM and ASAM variants are presented in the middle of the heatmap (white middle point). All models were trained with the default $\rho$. Blue represents an improvement in terms of validation accuracy over such baselines, while red indicates a degradation in performance.

Table 7: Generalization performance (validation accuracy %) of the different methods on several models trained on CIFAR-10 and CIFAR-100. The best and second-best methods are shaded in dark gray and gray, respectively.

| Dataset | CIFAR-10 | | CIFAR-100 | |
|---|---|---|---|---|
| Model | ResNet-34 | WRN-28-2 | ResNet-50 | WRN-28-10 |
| Multistep-SAM | $95.72_{\pm.15}$ | $94.39_{\pm.09}$ | $77.03_{\pm.65}$ | $80.55_{\pm.06}$ |
| + average grads | $95.74_{\pm.25}$ | $94.55_{\pm.22}$ | $76.97_{\pm.57}$ | $80.58_{\pm.21}$ |
| **Lookbehind + SAM** | $\mathbf{96.27_{\pm.07}}$ | $\mathbf{94.81_{\pm.22}}$ | $\mathbf{78.62_{\pm.48}}$ | $\mathbf{80.99_{\pm.02}}$ |
| **+ adaptive $\alpha$** | $\mathbf{96.33_{\pm.04}}$ | $\mathbf{94.88_{\pm.12}}$ | $\mathbf{78.33_{\pm.36}}$ | $\mathbf{80.86_{\pm.13}}$ |
| Multistep-ASAM | $95.91_{\pm.14}$ | $95.06_{\pm.15}$ | $77.81_{\pm.52}$ | $81.67_{\pm.06}$ |
| + average grads | $95.91_{\pm.24}$ | $94.92_{\pm.09}$ | $78.39_{\pm.52}$ | $81.35_{\pm.36}$ |
| **Lookbehind + ASAM** | $\mathbf{96.54_{\pm.21}}$ | $\mathbf{95.23_{\pm.01}}$ | $\mathbf{78.86_{\pm.29}}$ | $\mathbf{82.16_{\pm.09}}$ |
| **+ adaptive $\alpha$** | $\mathbf{96.57_{\pm.03}}$ | $\mathbf{95.08_{\pm.15}}$ | $\mathbf{78.89_{\pm.45}}$ | $\mathbf{81.86_{\pm.22}}$ |

## A.8 ADDITIONAL TRAINING SETUPS

To further illustrate the superiority of our approach with stronger baselines, we replicated the setup originally used by ASAM described in Kwon et al. (2021). The main difference between this new setup and our previous setup is the use of a cosine learning rate scheduler and label smoothing which leads to an increase in generalization performance. To avoid any hyperparameter tuning, we used $k = 2$ with an adaptive $\alpha$ for Lookbehind and used SAM and ASAM's default $\rho$ values of 0.1 and 1.0, respectively, as reported in Kwon et al. (2021).

Results over 5 seeds using a WRN-28-2 model trained on CIFAR-100 for 200 epochs are presented in Table 8. We observe that Lookbehind is able to further improve upon the high-accuracy SAM and ASAM baseline models. This supports our conclusions when using the setup used in the experiments in the main paper showcasing the superiority of our method.

Table 8: Generalization performance (test accuracy %) of the different methods on WRN-28-10 trained on CIFAR-100 using the same setup as ASAM. Results for SGD, SAM, and ASAM ($\star$) are the ones reported by Kwon et al. (2021).

| Model | WRN-28-10 |
|---|---|
| SGD$^\star$ | $81.56_{\pm.13}$ |
| SAM$^\star$ | $83.42_{\pm.04}$ |
| **Lookbehind + SAM** | $\mathbf{83.72_{\pm.10}}$ |
| ASAM$^\star$ | $83.68_{\pm.12}$ |
| **Lookbehind + ASAM** | $\mathbf{84.00_{\pm.18}}$ |

