# OpenReview forum: "Lookbehind Optimizer: k steps back, 1 step forward"
_ICLR.cc/2024/Conference — Submitted to ICLR 2024_

### Official Review · Reviewer_1aPq · 2023-10-27

**Soundness:** 3 good
**Presentation:** 3 good
**Contribution:** 2 fair
**Rating:** 3
**Confidence:** 4

**Summary:**

This paper proposes a new version of multi-step SAM using a linear interpolation technique. They discuss convergence properties of the algorithm and present numerical experiments.

**Strengths:**

The proposed method seems to outperform SAM in the settings studied.

**Weaknesses:**

- The idea of multi-step SAM is not new and has been explored before. Although the interpolation step makes this work different from the previous work, I think that is a marginal contribution.

- The numerical experiments are rather limited. As far as I could tell, they only consider CIFAR and a down-sized version of ImageNet, only using ResNets. I think the complete ImageNet should be in the numerical studies, and some transformer-based models need to be added (such as ViTs or BERT).

**Questions:**

NA

---

> ### Author Response · Authors · 2023-11-22
>
> We thank the reviewer for their comments. We believe there are a few misconceptions that we address below:
>
> > **“The idea of multi-step SAM is not new and has been explored before”**
>
> The primary aim of our work is not to introduce the concept of multiple ascent step SAM, but rather  to make it work well in practice. By enhancing SAM’s maximization and minimization steps, our approach demonstrated significant performance improvements over vanilla SAM and ASAM, as well as additional baseline methods.
>
> > **“Although the interpolation step makes this work different from the previous work, I think that is a marginal contribution.”**
>
> Beyond (gradient combination and)  linear interpolation, we also introduce a mechanism to analytically control $\alpha$ during training (**Section 6**). By considering the angle between the first gathered gradient and the final update direction, we successfully remove the need to finetune $\alpha$, while maintaining performance. We show that our method leads to improved performance in a myriad of tasks: generalization performance, model robustness, and lifelong learning. We believe these contributions are substantial rather than marginal.
>
> > **“The complete ImageNet should be in the numerical studies”**
>
> We want to clarify that our experiments used ImageNet-1k with an image size of 224x224, following common practice in the literature. We added this clarification in **Appendix A.3** in the manuscript.
>
> > **“Transformer-based models need to be added”**
>
> SAM has been successfully applied to Transformers [1] and we should expect Lookbehind to further improve SAM’s performance, as presented throughout our paper. While we understand the current desire to use Transformers in the community, we do not think this is a critical issue with our manuscript that warrants a reason for rejection.
>
> [1] Chen et al. When Vision Transformers Outperform ResNets without Pre-training or Strong Data Augmentations. ICLR 2022.

---

> > ### Comment · Reviewer_1aPq · 2023-11-22
> >
> > Thank you for your rebuttal. I will keep my evaluation.

---

### Official Review · Reviewer_aoaQ · 2023-10-31

**Soundness:** 2 fair
**Presentation:** 2 fair
**Contribution:** 2 fair
**Rating:** 3
**Confidence:** 4

**Summary:**

The authors focuses on contributing to the Sharpness-Aware Minimization (SAM) algorithm, more specifically the multiple-ascent SAM, where they proposed Lookbehind SAM. Lookbehind SAM average the history gradient solved during multiple ascent steps for each batch training. e authors empirically show the effectiveness of their method via experiments on CIFAR, ImageNet and as well as lifelong learning settings.

**Strengths:**

1. The paper is clearly written and easy to follow. But some of the presentations need to be improved.
2. The proposed method seems interesting.
3. I notice that the authors present valuable extra results, i.e. lifelong learning, in addition to the image classification that is typically used in SAM.

**Weaknesses:**

1. My first concern may be the title this paper, "Lookbehind Optimizer: k steps back, 1 step forward". The title obviously mimics the paper "Lookahead Optimizer: k steps forward, 1 step back". This somehow directly implies that the method of the presented paper is opposite to that in Lookahead paper. But they are very different. The proposed Lookbehind could not be used without SAM, and focusing on different gradients. Considering the proposed method contributes specifically to SAM, so at least, the paper ought to mention SAM algorithm in their title to give a clear view, such as Lookbehind SAM Optimizer or Lookbehind SAM. Not mentioning SAM in the title is unacceptable.

2. The motivation is not quite clear for me. The authors claim that the proposed method could reduce the variance derived from the multiple ascent steps. Why we need to reduce such variance is not clearly demonstrated.

3. Although the proposed method seems interesting, yet its drawback is quite obvious. A single parameter update requires performing multiple backwards propagation to calculate the ascent gradient. In other words, the utility efficiency of training samples is quite low. Multiple backwards propagations are required on the same sample batch. Given that the improvement of the proposed method is marginal, the side effect of such a method can potentially be substantial. So, in my opinion, the proposed method may not be better than the vanilla SAM.

4. Based on my tuning experience, the authors have not trained the models to achieve comparable results when using vanilla SAM in their baseline on Cifar dataset. The authors should at least trained models to achieve comparable results with those reported in the original SAM paper. For example, in the original SAM paper, WRN-28-10 can achieve an error rate of 16.5 with SAM, while in the given paper, it achieves an error rate of 19.5 with SAM, almost 3 percentage gap. Therefore, the reported results can not fully persuade me that their method is more effective. Also, many advanced SAM variants have not mentioned in their experiments. It is highly recommended that the authors make comparisons with these SOTA methods.

5. For Figure 2, it is recommended that the authors draw some marks with respect to the gradient vector.

6. For Algm 1, it is recommended that the authors use $\theta$ to substitute fast weights $\phi$, as that used in the Lookahead paper.

**Questions:**

See Weakness.

**Details Of Ethics Concerns:**

I have not found any discussions about the limitations and potential negative societal impact. But in my opinion, this may not be a problem, since the work only focuses on the learning method in machine learning. Still, it is highly encouraged to add corresponding discussions.

---

> ### Author Response · Authors · 2023-11-22
>
> We thank the reviewer for their constructive feedback and concrete suggestions to better articulate our motivation and contribution. Below we explain how we address the reviewer’s concerns.
>
> >  **“The paper ought to mention SAM algorithm in their title to give a clear view, such as Lookbehind SAM Optimizer or Lookbehind SAM.”**
>
> Thank you for bringing attention to this oversight. Initially, our choice of title was influenced by the parallel with Lookahead, aiming for consistency in titles. However, we acknowledge the merit of including SAM explicitly for a clearer representation of the paper's content. In response to this valuable feedback, we have updated the title to **"Lookbehind-SAM: k steps back, 1 step forward."** This modification aims to maintain brevity while ensuring a direct connection with the SAM algorithm. We appreciate the suggestion and believe this title better conveys the essence of our work.
>
> > **“The authors claim that the proposed method could reduce the variance derived from the multiple ascent steps. Why we need to reduce such variance is not clearly demonstrated.”**
>
> We thank the reviewer for this opportunity to clarify this important point. In the context of Multistep-SAM, the objective is to enhance the solution of the maximization part of SAM’s objective. However, the substantial departure of the gradient collected far away from the original point can adversely impact the original minimization objective, resulting in a suboptimal loss-sharpness trade-off, as illustrated in **Figure 1**.
>
> Lookbehind addresses this issue by collecting and combining gradients along the ascent step, effectively reducing the variance of these gradients. This reduction in variance contributes to an improved loss-sharpness trade-off, ultimately leading to enhanced performance. We have incorporated additional explanations and clarifications in **Section 3** of the manuscript to further elucidate this critical aspect. We hope this provides a clearer understanding of the motivation behind our approach.
>
> > **“A single parameter update requires performing multiple backwards propagation to calculate the ascent gradient. In other words, the utility efficiency of training samples is quite low”**
>
> We agree with the reviewer’s concern about the increased computational overhead associated with multiple backwards propagation for a single parameter update, which in fact applies to any Multistep Sam method. Note that Lookbehind entails the same number of gradient computations as Multistep-SAM/ASAM, yet consistently surpasses the latter in performance. Through this lens, although the utility efficiency of training samples is the same for both methods, the utility efficacy is higher with Lookbehind.
>
> > **“the authors have not trained the models to achieve comparable results when using vanilla SAM in their baseline on Cifar dataset.”**
>
> We appreciate your concern about the strength of the baselines and have taken steps to address it. We conducted additional experiments in line with the training setup from ASAM's paper [1], by applying a cosine learning rate scheduler and label smoothing to WRN-28-10 trained on CIFAR-100. The results, detailed in **Appendix A.8 (Table 8)**, demonstrate that Lookbehind achieves an error rate of $16.28$ and $16.00$ for SAM and ASAM, respectively. These performances convincingly outperform the error rates of $16.58$ and $16.32$ reported for vanilla SAM and ASAM in [1]. We hope these additional results provide a more robust basis for comparison.
>
> [1] Kwon et al. ASAM: Adaptive Sharpness-Aware Minimization for Scale-Invariant Learning of Deep Neural Networks. ICML 2021.
>
> > **“Many advanced SAM variants have not mentioned in their experiments”**
>
> Thank you for your suggestion. It’s important to note that Lookbehind is a versatile wrapper that can be used with any SAM variant. While our focus has been on the original SAM algorithm and the adaptive variant (ASAM), our results indicate that Lookbehind can be applied to further improve the loss-sharpness trade-off and generalization performance across a range of SAM variants.

---

> > ### Comment · Reviewer_aoaQ · 2023-11-22
> > **Thanks for the kind response.**
> >
> > I have read the response, and thanks for the effort. The authors have addressed my concerns to an extent. However, the proposed method natually suffers from low computing efficiency. And the authors have not compared with enough contempory works.  Therefore, I would like to raise my score to weak reject.

---

> ### Author Response · Authors · 2023-11-22
>
> Thank you for your response and the consideration of the score adjustment. We kindly note that the reviewer’s original score of 3 is still reflected on Openreview, and we appreciate your attention to this matter.
>
> Regarding the concern about computational overhead, we recognize its significance, but it’s essential to clarify that this issue is inherent to any multiple ascent-step SAM algorithm, not specific to our method. Our paper contributes to the active research on multiple ascent-step SAM, focusing on addressing the performance challenge arising from a poor sharpness-loss tradeoff. In response to the reviewer’s feedback, we’ve added a note in the conclusion, underscoring the importance of exploring the computational efficiency of multistep SAM.
>
> While testing Lookbehind with a broader range of SAM variants is an interesting suggestion,  it’s crucial to note that Lookbehind is not just a new SAM variant. Instead, it serves as a versatile wrapper applicable to any sharpness-aware minimization method, aiming to enhance their sharpness-loss tradeoff.  We firmly believe our results highlight the effectiveness of  Lookbehind in this context.

---

### Official Review · Reviewer_tnTm · 2023-11-01

**Soundness:** 3 good
**Presentation:** 3 good
**Contribution:** 2 fair
**Rating:** 5
**Confidence:** 4

**Summary:**

The paper presents a new approach for the multiple ascent steps Sharpness-aware minimization training, which has been proven to enhance the generalization ability of neural networks. In particular, the authors introduce a variance reduction technique to better leverage the information along the trajectory instead of using the last updated model parameter only. Extensive experiment results on both single-task and continual learning show the effectiveness of the proposed approach.

**Strengths:**

- While previous studies have shown that multi-step ascent SAM does not improve over single-step SAM, the paper proposes to utilize multiple gradients along the ascent trajectory for a better maximization step. Motivated by the Lookahead optimizer, the proposed method stabilizes the training, thus improving the performance of the model.
- The paper is well-written and easy to follow.
- The authors conduct experiments on many datasets and backbones and empirically verify the benefit of Lookbehind SAM over SAM. The ablation studies showcase how their method is better than naive Multistep-SAM and Lookahead SAM.
- Lookbehind is readily applicable to different SAM-based training methods (e.g. ASAM). Moreover, it is robust to the hyperparameter tuning.
- The proposed adaptive $alpha$ utilizes the similarity in the updating directions between the first and last gradients can eliminate the need to tune this parameter while maintaining superior performance.

**Weaknesses:**

- The authors claim that "a drawback of any multiple ascent step SAM method is the computational overhead which increases training time by a factor k". However, while Lookbehind calculates k ascent steps (line 6 in Algorithm 1) and k descent steps (line 8), Multistep-SAM performs k ascent steps and only a single descent step, requiring almost half the complexity.

- Can the authors compare Lookbehind against averaging the ascent gradients baseline, which has been proven to be able to improve SAM [1] (and also performs k ascent steps and a single descent step only)?

- More detailed descriptions are needed for Figure 2 and Figure 11. The decay term can be omitted for simplification.

[1] Kim, Hoki, et al. "Exploring the effect of multi-step ascent in sharpness-aware minimization." arXiv preprint arXiv:2302.10181 (2023).

**Questions:**

- While leveraging multiple ascent steps can improve over the original SAM/ASAM, a prior study [2] shows that the inner gradient ascent can be calculated periodically while maintaining similar performance to the conventional SAM (i.e. it is redundant to compute the ascent gradient at every step). Can the author elaborate more on this?

- Since the computational complexity is multiplied by k, can the authors compare Lookbehind against SAM/ASAM at different training budgets?

[2] Liu, Yong, et al. "Towards efficient and scalable sharpness-aware minimization." Proceedings of the IEEE/CVF Conference on Computer Vision and Pattern Recognition. 2022.

---

> ### Author Response · Authors · 2023-11-22
>
> We thank the reviewer for their accurate summary of our work. We also thank the reviewer for their questions and concrete suggestions to clarify our contribution and to improve both  the content and presentation.
>
> > **“Can the authors compare Lookbehind against averaging the ascent gradients baseline ?”**
>
> Thank you for this suggestion. We have added additional comparisons between Lookbehind and Multistep-SAM/ASAM with averaging the ascent gradients. Results are shown in **Appendix A7 (Table 7)** of the updated manuscript. We observe that despite showing some improvements over Multistep-SAM/ASAM, both variants of Lookbehind (with and without adaptive $\alpha$) outperform these baselines.
>
> > **“More detailed descriptions are needed for Figure 2 and Figure 11.”**
>
> Thank you for pointing this out. We added a detailed description of Figure 11 and presented the key differences between Figures 2 and 11 in **Appendix A.2**.
>
> > **“The decay term can be omitted for simplification.”**
>
> Thank you for this suggestion. We removed the decay term from all equations and kept the mention of using weight decay in **Appendix A.3** where we describe the training details.
>
> > **“While leveraging multiple ascent steps can improve over the original SAM/ASAM, a prior study [2] shows that the inner gradient ascent can be calculated periodically while maintaining similar performance to the conventional SAM (i.e. it is redundant to compute the ascent gradient at every step). Can the author elaborate more on this?”**
>
> Lookbehind is a general wrapper for any sharpness-aware minimization method. Therefore, more efficient SAM variants such as re-using the inner gradient ascent steps are orthogonal to our work and may be used together to improve training efficiency.
>
> > **“Can the authors compare Lookbehind against SAM/ASAM at different training budgets?”**
>
> We understand and share the reviewer’s concern. Indeed, we already briefly discuss this in **Appendix A.1.1**, where we study the average number of epochs at which SAM and ASAM reach top validation performance. Our findings suggest that increasing training time would not be a viable way to further improve performance in our setup.
>
> In response to the reviewer’s recommendation,  we conducted additional experiments by replicating the ASAM setup. We trained a WRN-28-10 model on CIFAR-100 using SAM and ASAM for an extended period of  400 epochs instead of 200. The results, detailed in Appendix A.8, Table 8, show that the 400-epoch models achieved similar generalization performance compared to the original 200 epochs. Specifically, SAM and ASAM attained  $83.37_{\pm .12}$ and $83.67_{\pm .12}$ validation accuracy at an average epoch of $194.8_{\pm 2.78}$ and $202.8_{\pm 8.23}$, respectively. This supports our earlier observation that the models’ generalization performance was already saturated by the original 200 epochs. We hope this additional analysis addresses the reviewer’s concern.

---

### Official Review · Reviewer_sgRK · 2023-11-05

**Soundness:** 4 excellent
**Presentation:** 4 excellent
**Contribution:** 3 good
**Rating:** 8
**Confidence:** 4

**Summary:**

The authors propose a novel optimization method, called Lookbehind, that leverages the benefits of multiple ascent steps and linear interpolation to improve the efficiency of the maximization and minimization parts of sharpness-aware minimization (SAM). The experiments show that Lookbehind improves the generalization performance across various models and datasets, increases model robustness, and promotes the ability to continuously learn in lifelong learning settings.

**Strengths:**

S1. The work is well motivated. Finding a simple but effective method to improve SAM is both interesting and important.

S2. The paper is well written and easy to follow. The illustration in Fig. 1 and Fig. 2 are helpful to understand the main results.

S3. The authors conduct numerous experiments to showcase the benefits of achieving a better sharpness-loss trade-off in SAM methods. These experiments are comprehensive and convincing. Additionally, the paper includes several ablation studies.

**Weaknesses:**

W1. As mentioned by the authors, one inherent drawback of Lookbehind is the computational overhead, which leads to an increase in training time by a factor of $k$.

W2. No convergence analysis for Lookbehind is provided.

**Questions:**

Q1. Drawing inspiration from the Lookahead optimizer, can the fast weights (updated in line 8 of Algorithm 1) be approximately updated using any standard optimization algorithm like SGD or Adam?

Q2. Why not conduct an analysis of the sensitivity of Lookbehind to the step size $\eta$ for the fast weights?

Q3. Anderson acceleration has a similar flavor to Lookbehind, and it has been employed in solving minimax optimization problems.What is the relation between Lookbehind and Anderson acceleration?

Minor Comments:

(1) On page 3, in line 4 from below, should "slow weights" be replaced with "fast weights"?

(2) On page 8, in line 13 from below, should "$\rho$” be repaced with "$\alpha$”?

(3) On page 8, in line 4 from below, "$0\geq \alpha^*<1$” should be "$0\leq \alpha^*<1$”.

---

> ### Author Response · Authors · 2023-11-22
>
> Thank you for your comments. We are grateful for your positive feedback and overall excitement regarding our work. We address specific questions below:
>
> > **“Can the fast weights (updated in line 8 of Algorithm 1) be approximately updated using any standard optimization algorithm like SGD or Adam?”**
>
> Thank you for this important question. Yes, the fast weights updated in line 8 of Algorithm 1 can be approximately updated using standard optimization algorithms like SGD or Adam.
>  Lookbehind is a wrapper for sharpness-aware minimization methods, and these, in turn,  are a wrapper for standard optimization algorithms like SGD or Adam. Although SAM and ASAM have been primarily used with SGD, both methods have also been successfully combined with Adam, showing improvements over vanilla Adam [1].
>
> [1] Kwon et al. ASAM: Adaptive Sharpness-Aware Minimization for Scale-Invariant Learning of Deep Neural Networks. ICML 2021.
>
> > **“Why not conduct an analysis of the sensitivity of Lookbehind to the step size of the fast weights?”**
>
> We appreciate your suggestion regarding the analysis of Lookbehind's sensitivity to the step size of the fast weights. However, we maintain a consistent inner learning rate for all methods. In the context of our specific method, the influence of the inner step size can be managed through the parameter $\alpha$, which can be considered as compensation for variations in the learning rate applied to the fast weights.
>
>
> > **“What is the relation between Lookbehind and Anderson acceleration?”**
>
>
> Thank you for bringing up this interesting question. As far as we understand, the goals of Lookbehind and Anderson acceleration appear to be distinct. Anderson acceleration is typically employed as a convergence acceleration method, whereas Lookbehind serves as a variance-reduction method, enhancing the efficiency of the maximization and minimization components of SAM’s objective. However, we acknowledge that we were not familiar with Anderson acceleration before this discussion and are open to further conversations on this topic.
>
> **Minor fixes**: We changed the text where appropriate to address your comments - thank you for your corrections.

---

> > ### Comment · Reviewer_sgRK · 2023-11-22
> >
> > Thanks for the kind response. The authors have addressed my concerns.

---

### Author Response · Authors · 2023-11-22
**General response**

We are grateful to all reviewers for their engaged feedback and meaningful comments. We are happy to see that they found our soundness and presentation to be excellent (**Reviewer sgRK**) and good (**Reviewers tnTm, 1aPq**). We are encouraged they found our work to be well-motivated and important (**Reviewer sgRK**), our paper to be well-written and easy to follow (**Reviewers sgRK, tnTm, aoaQ**), and our method to be interesting (**Reviewers sgRK, aoaQ**). Finally, we appreciate the recognition of our experiments being comprehensive (**Reviewers sgRK, tnTm**) and convincing (**Reviewer sgRK**), while having extensive ablations (**Reviewers sgRK, tnTm**).

We have worked to address the reviewers’ feedback and incorporate their suggestions in our manuscript (changes highlighted in blue). The main modifications may be summarized as follows:
- We changed the title to “**Lookbehind-SAM: $k$ steps back, 1 step forward**”, following **Reviewer aoaQ**’s recommendation to include SAM in the title.
- We added a new section (**Appendix A.7**) with a comparison between our method and Multistep-SAM/ASAM with gradient averaging, as requested by **Reviewer tnTm**. We observe that Lookbehind always outperforms this additional baseline.
- We included a new section (**Appendix A.8**) with a comparison between our method and SAM/ASAM using a stronger baseline WRN-28-10 model, as requested by **Reviewer aoaQ**. We observe that Lookbehind outperforms the previously reported generalization results for SAM and ASAM when using the same training setup as ASAM’s paper.
- Finally, we included additional ImageNet experiments using adaptive $\alpha$ to further showcase to avoid the need to treat $\alpha$ as a hyperparameter (**Table 3**). We would like to highlight that **both variants of Lookbehind (with and without an adaptive $\alpha$) outperform all the other methods** on all tested models and datasets (c.f. **Table 2**). We hope this further highlights our method’s overall contribution.

We address each reviewer individually to clarify additional points and promote further discussions. We would like to thank the reviewers once again for their valuable feedback.

---

### Meta-Review · Area_Chair_86AK · 2023-12-22

**Metareview:**

The paper proposes a method they call Lookbehind-SAM. It is targetted at achieving a better training loss sharpness tradeoff than SAM or SAM like methods. The main proposal is to perform multiple ascent steps in the inner loop of SAM and accumulate the gradient ateach of those inner steps towards identifying a descent direction (SAM will do the same thing but with 1 step). Finally they further employ an idea from a Lookahead optimizer where in after accumulating these gradients you bias the final point a little towards the starting point of the inner loop. The paper's main contributions are making this proposal and then they perform quite extensive experiments towards showing that their proposal leads to better training/sharpness tradeoff when compared with SAM or a multistep ascent version of SAM.

There are many things to like about the investigation conducted in the paper -- primarily the well considered ablation of their two ideas of multiple ascent and the look ahead part. They verified both these ideas in isolation with SAM as well. The experiments are on reasonable datasets and the authors have provided many experiments on the effects of various parameters. However there are a few things that make the method a significant contribution in practice. The paper should primarily be seen as how to to multi ascent step SAM more effectively and the two ideas they introduce eventually are targetted towards it. The main question raised by the reviewers evetntually are whether multi-ascent steps SAM is itself worth while in practice due to its siginificant computational inefficiency. Further the central idea of the paper is a combination of two relatively simple ideas one of them Lookahead is a generic wrapper over any algorithms. So in this sense the reviewers find the novelty of the contribution limited as well.

Overall due to above reasons multiple reviewers and the meta reviewer find the paper slightly below the borderline of acceptance at ICLR. Nevertheless I sincerely commend the authors on their methodical and well though out and presented investigation. This is a rarity.

**Justification For Why Not Higher Score:**

Highlighted in the meta review. The core reason is the limited potential impact of the paper's proposal due to its inherent inefficiency. The limited novelty in terms of ideas and tools introduced is another limiting aspect.

**Justification For Why Not Lower Score:**

NA

---

### Decision · Program_Chairs · 2024-01-16

Reject